# NVIDIA FLARE:
# Federated Learning from Simulation to Real-World

**Holger R. Roth    Yan Cheng    Yuhong Wen    Isaac Yang    Ziyue Xu    Yuan-Ting Hsieh**
**Kristopher Kersten    Ahmed Harouni    Can Zhao    Kevin Lu    Zhihong Zhang    Wenqi Li**
**Andriy Myronenko    Dong Yang    Sean Yang    Nicola Rieke    Abood Quraini    Chester Chen**
**Daguang Xu    Nic Ma    Prerna Dogra    Mona Flores    Andrew Feng**

NVIDIA Corporation*
Shanghai, China
Munich, Germany
Bethesda, Santa Clara, USA

## Abstract

Federated learning (FL) enables the building of robust and generalizable AI models by leveraging diverse datasets from multiple collaborators without centralizing the data. We created NVIDIA FLARE[1] as an open-source software development kit (SDK) to make it easier for data scientists to use FL in their research and real-world applications. The SDK includes solutions for state-of-the-art FL algorithms and federated machine learning approaches, which facilitate building workflows for distributed learning across enterprises and enable platform developers to create a secure, privacy-preserving offering for multiparty collaboration utilizing homomorphic encryption or differential privacy. The SDK is a lightweight, flexible, and scalable Python package, and allows researchers to bring their data science workflows implemented in any training libraries (PyTorch, TensorFlow, XGBoost, or even NumPy) and apply them in real-world FL settings. This paper introduces the key design principles of FLARE and illustrates some use cases (e.g., COVID analysis) with customizable FL workflows that implement different privacy-preserving algorithms.

## 1   Introduction

Federated learning (FL) has become a reality for many real-world applications [22]. It enables multinational collaborations on a global scale to build more robust and generalizable machine learning and AI models. In this paper, we introduce NVIDIA FLARE, an open-source software development kit (SDK) that makes it easier for data scientists to collaborate to develop more generalizable and robust AI models by sharing model weights rather than private data. While FL is attractive in many industries, it is particularly beneficial for healthcare applications where patient data needs to be protected. For example, FL has been used for predicting clinical outcomes in patients with COVID-19 [6] or to segment brain lesions in magnetic resonance imaging [26, 25]. FLARE is not limited to applications in healthcare and is designed to allow cross-silo FL [11] across enterprises for different industries and researchers.

In recent years, several efforts (both open-source and commercial) have been made to bring FL technology into the healthcare sector and other industries, like TensorFlow Federated [1], PySyft [32], FedML [10], FATE [16], Flower [2], OpenFL [21], Fed-BioMed [27], IBM Federated Learning [17], HP Swarm Learning [29], FederatedScope [30], FLUTE [7], and more. Some focus on simulated FL settings for researchers, while others prioritize production settings. FLARE aims to be useful for both sce-

---

*Contact: {hroth,yanc,chesterc,daguangx,pdogra,andyf}@nvidia.com
[1]Code is available at https://github.com/NVIDIA/NVFlare.

36th Conference on Neural Information Processing Systems (NeurIPS 2022).

narios: 1) for researchers by providing efficient and extensible simulation tools and 2) by providing an easy path to transfer research into real-world production settings, supporting high availability and server failover, and by providing additional productivity tools such as multi-tasking and admin commands.

## 2 NVIDIA FLARE Overview

FLARE stands for "Federated Learning Application Runtime Environment". The SDK enables researchers and data scientists to adapt their existing machine learning and deep learning workflows to a federated paradigm and enables platform developers to build a secure, privacy-preserving offering for distributed multiparty collaboration.

FLARE is a lightweight, flexible, and scalable federated learning framework implemented in Python that is agnostic to the underlying training library. Developers can bring their own data science workflows implemented in PyTorch, TensorFlow, or even in pure NumPy, and apply them in a federated setting. A typical FL workflow such as the popular federated averaging (FedAvg) algorithm [18], can be implemented in FLARE using the following main steps. Starting from an initial global model, each FL client trains the model on their local data for a certain amount of time and sends model updates to the server for aggregation. The server then uses the aggregated updates to update the global model for the next round of training. This process is iterated many times until the model converges.

Though used heavily for federated deep learning, FLARE is a generic approach for supporting collaborative computing across multiple clients. FLARE provides the *Controller* programming API for researchers to create workflows for coordinating clients for the purpose of collaboration. FedAvg is one such workflow. Another example is cyclic weight transfer [4]. The central concept of collaboration is the notion of "task". An FL controller assigns tasks (e.g., deep-learning training with model weights) to one or more FL clients and processes results returned from clients (e.g., model weight updates). The controller may assign additional tasks to clients based on the processed results and other factors (e.g., a pre-configured number of training rounds). This task-based interaction continues until the objectives of the study are achieved.

The API supports typical controller-client interaction patterns like broadcasting a task to multiple clients, sending a task to one or more specified clients, or relaying a task to multiple clients sequentially. Each interaction pattern comes with two flavors: wait (block until results from clients are received) or no-wait. A workflow developer can use any of these interaction patterns to create innovative workflows. For example, the *ScatterAndGather* controller (typically used for FedAvg-like algorithms) is implemented with the *broadcast_and_wait* pattern, and the *CyclicController* is implemented with the *relay_and_wait pattern*. The controller API allows the researcher to focus on the control logic without needing to deal with underlying communication issues. Figure 1 shows the principle. Each FL client acts as a worker that simply executes tasks assigned to it (e.g., model training) and returns execution results to the controller. At each task interaction, there can be optional filters that process the task data or results before passing it to the *Controller* (on the server side) or task executor (client side). The filter mechanism can be used for data privacy protection (e.g., homomorphic encryption/decryption or differential privacy) without having to alter the training algorithms.

**Key Components**   NVIDIA FLARE is built on a componentized architecture that gives the flexibility to take FL workloads from research and simulation to real-world production deployment. Some of the key components of this SDK include:

- **FL Simulator** for rapid development and prototyping.
- **FLARE Dashboard** for simplified project management, secure provisioning, and deployment, orchestration.
- **Reference FL algorithms** (e.g., FedAvg, FedProx, SCAFFOLD) and workflows (e.g., Scatter and Gather, Cyclic).
- **Privacy preservation** with differential privacy, homomorphic encryption, and more.
- **Specification-based API** for extensibility, allowing customization with plug-able components.
- **Tight integration** with other learning frameworks like MONAI [3], XGBoost [5], and more.

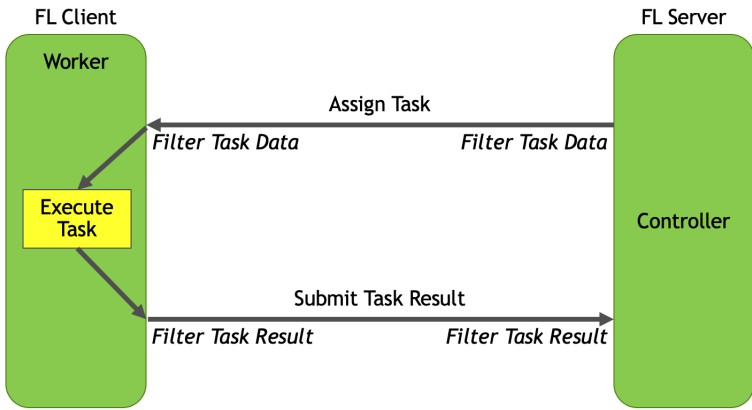

Figure 1: FLARE job execution. The *Controller* is a Python object that controls or coordinates the *Workers* to get a job done. The controller is run on the FL server. A *Worker* is capable of performing tasks. *Workers* run on FL clients.

**High-Level Architecture**  FLARE is designed with the idea that less is more, using a specification-based design principle to focus on what is essential. This allows other people to be able to do what they want to do in real-world applications by following clear API definitions. FL is an open-ended space.  The API-based design allows others to bring their implementations and solutions for various components.  Controllers, task executors, and filters are just examples of such extensible components. FLARE provides an end-to-end operation environment for different personas. It provides a comprehensive provisioning system that creates security credentials for secure communications to enable the easy and secure deployment of FL applications in the real world.  It also provides an FL Simulator for running proof-of-concept studies locally.  In production mode, the researcher conducts an FL study by submitting jobs using admin commands either on Notebook or FLARE Console – an interactive command tool. FLARE provides many commands for system operation and job management. With these commands, one can start and stop a specific client or the entire system, submit new jobs, check the status of jobs, create a job by cloning from an existing one, and much more.

With FLARE's component-based design, a job is just a configuration of components needed for the study. For the control logic, the job specifies the controller component to be used and any components required by the controller.

## 3    System Concepts

A FLARE system is a typical client-server communication system that comprises one or more FL server(s), one or more FL client(s), and one or more admin clients. The FL Servers open two ports for communication with FL clients and admin clients. FL clients and admin clients connect to the opened ports. FL clients and admin clients do not open any ports and do not directly communicate with each other.  The following is an overview of the key concepts and objects available in FLARE and the information that can be passed between them.

**Workers and Controller**  FLARE's collaborative computing is achieved through the *Controller/Worker* interactions.

**Shareable**  Object that represents a communication between server and client. Technically, the *Shareable* is implemented as a Python dictionary that could contain different information, e.g., model weights.

**Data Exchange Object (DXO)**  Standardizes the data passed between the communicating parties. One can think of the *Shareable* as the envelope and the *DXO* as the letter. Together, they comprise a message to be shared between communicating parties.

**FLComponent**   The base class of all the FL components. Executors, controllers, filters, aggregators, and their subtypes are all *FLComponents*. *FLComponent* comes with some useful built-in methods for logging, event handling, auditing, and error handling.

**Executors**   Type of *FLComponent* for FL clients that has an execute method that produces a *Shareable* from an input *Shareable*. FLARE provides both single- and multi-process executors to implement different computing workloads.

**FLContext**   One of the most important features of FLARE is to pass data between the FL components. *FLContext* is available to every method of all common FLComponent types. Through *FLContext*, the component developer can get services provided by the underlying infrastructure and share data with other components of the FL system.

**Filters**   Filters in FLARE are a type of *FLComponent* that have a process method to transform the *Shareable* object between the communicating parties. A Filter can be used to provide additional processing to shareable data before sending or after receiving from a peer. Filters can convert data formats and a lot more and are FLARE's primary mechanism for data privacy protection [15, 9]:

- *ExcludeVars* to exclude variables from shareable.
- *PercentilePrivacy* for truncation of weights by percentile.
- *SVTPrivacy* for differential privacy through sparse vector techniques.
- Homomorphic encryption filters used for secure aggregation.

As an example, we show the average encryption, decryption, and upload times when using homomorphic encryption for secure aggregation[2]. We compare raw data to encrypted model gradients uploaded in Fig. 2 and Table 1 when hosting the server on AWS[3] and connecting 30 clients. One can see the longer upload times due to the larger message sizes needed by homomorphic encryption.

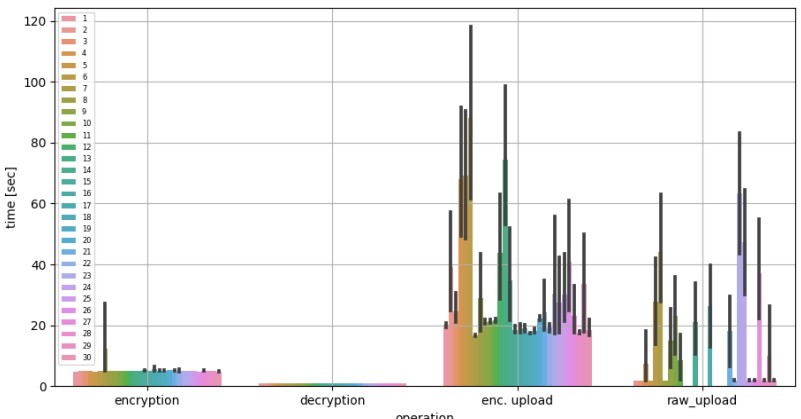

Figure 2: Running federated learning with 30 clients and the server on AWS.

Table 1: Federated learning exchanging homomorphic encrypted vs. g raw model updates.

| Time in seconds | Mean | Std. Dev. |
|---|---|---|
| Encryption | 5.01 | 1.18 |
| Decryption | 0.95 | 0.04 |
| Enc. upload | 38.00 | 71.17 |
| Raw upload | 21.57 | 74.23 |

---

[2]https://developer.nvidia.com/blog/federated-learning-with-homomorphic-encryption
[3]For reference, we used an m5a.2xlarge instance with eight vCPUs, 32-GB memory, and up to 2,880 Gbps network bandwidth.

**Event Mechanism** FLARE comes with a powerful event mechanism that allows dynamic notifications to be sent to all event handlers. This mechanism enables data-based communication among decoupled components: one component fires an event when a certain condition occurs, and other components can listen to that event and processes the event data. Each *FLComponent* is automatically an event handler. To listen to and process an event, one can simply implement the *handle_event()* method and process desired event types. Events represent some important moments during the execution of the system logic. For example, before and after aggregation or when important data becomes available, e.g., a new "best" model was selected.

## 3.1 Productivity Features

FLARE contains features that enable efficient, collaborative, and robust computing workflows.

**Multi-tasking** For systems with a large capacity, computing resources could be idle most of the time. FLARE implements a resource-based multi-tasking solution, where multiple jobs can be run concurrently when overall system resources are available. Multi-tasking is made possible by a job scheduler on the server side that constantly tries to schedule a new job. For each job to be scheduled, the scheduler asks each client whether they can satisfy the required resources of the job (e.g., number of GPU devices) by querying the client's resource manager. If all clients can meet the requirement, the job will be scheduled and deployed to the clients.

**High Availability and Server Failover** To avoid the FL server as a single point of failure, a solution has been implemented to support multiple FL servers with automatic cut-over when the currently active server becomes unavailable. Therefore, a component called *Overseer* is added to facilitate automatic cut-over. The *Overseer* provides the authoritative endpoint info of the active FL server. All other system entities (FL servers, FL clients, admin clients) constantly communicate (i.e., every 5 seconds) with the Overseer to obtain such information and act on it. If the server cutover happens during the execution of a job, then the job will continue to run on the new server. Depending on how the controller is written, the job may or may not need to restart from the beginning but can continue from a previously saved snapshot.

**Simulator** FLARE provides a simulator to allow data scientists and system developers to easily write new *FLComponents* and novel workflows. The simulator is a command line tool to run a FLARE job. To allow simple experimentation and debugging, the FL server and multiple clients run in the same process during simulation. A multi-process option allows making efficient use of resources, e.g., training multiple clients on different GPUs. The simulator follows the same job execution as in real-world FLARE deployment. Therefore, components developed in simulation can be directly deployed in real-world federated scenarios.

## 3.2 Secure Provisioning in FLARE

Security is an important requirement for federated learning systems. FLARE provides security solutions in the following areas: authentication, communication confidentiality, user authorization, data privacy protection, auditing, and local client policies.

**Authentication** FLARE ensures the identities of communicating peers with the use of mutual Transport Layer Security (TLS). Each participating party (FL Servers, Overseer, FL Clients, Admin Clients) must be properly provisioned. Once provisioned, each party receives a startup kit, which contains TLS credentials (public cert of the root, the party's own private key and certificate) and system endpoint information, see Fig. 3. Each party can only connect to the FLARE system with the startup kit. Communication confidentiality is also achieved with the use of TLS-based messaging.

**Federated Authorization** FLARE's admin command system is very rich and powerful. Not every command is for everyone. FLARE implements a role-based user authorization system that controls what a user can or cannot do. At the time of provision, each user is assigned a role. Authorization policies specify which commands are permitted for which roles. Each FL client can define its own authorization policy that specifies what a role can or cannot do to the client. For example, one client could allow a role to run jobs from any researchers, whereas another client may only allow jobs submitted by its own researchers (i.e., the client and the job submitter belong to the same organization).

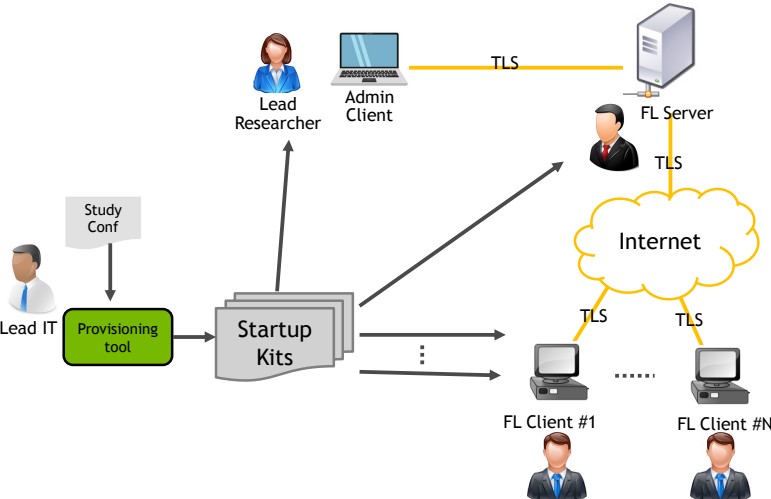

Figure 3: High-level steps for running a real-world study with secure provisioning with FLARE.

FLARE automatically records all user commands and job events in system audit files on both the server and client sides. In addition, the audit API can be used by application developers to record additional events in the audit files.

**Client-Privacy**  FLARE enhances the overall system security by allowing each client to define its own policies for authorization, data privacy (filters), as well as computing resource management. The client can change its policies at any time after the system is up and running without having to be re-provisioned. For example, the client could require that all jobs running on it are subject to a set of filters. The client could also change the number of computing resources (e.g., GPU devices) to be used by the FL client.

**Federated Data Science**  As a general distributed computing platform, FLARE can be used for various applications in different industries. Here we describe some of the most common use cases where FLARE was deployed so far.

**Federated Learning**  A go-to example dataset for benchmarking different FL algorithms is CIFAR-10. FLARE allows users to experiment with different algorithms and data splits using different levels of heterogeneity based on a Dirichlet sampling strategy [28]. Figure 4a shows the impact of different alpha values, where lower values cause higher heterogeneity on the performance of the FedAvg.

Apart from FedAvg, currently available in FLARE include FedProx [14], FedOpt [20], and SCAFFOLD [12]. Figure 4b compares an $\alpha$ setting of 0.1, causing a high data heterogeneity across clients and its impact on more advanced FL algorithms, namely FedProx, FedOpt, and SCAFFOLD. FedOpt and SCAFFOLD show markedly better convergence rates and achieve better performance than FedAvg and FedProx with the same alpha setting. SCAFFOLD achieves this by adding a correction term when updating the client models, while FedOpt utilizes SGD with momentum to update the global model on the server. Therefore, both achieve better performance with the same number of training steps as FedAvg and FedProx.

Other algorithms available in or coming soon to FLARE include federated XGBoost [5], Ditto [13], FedSM [31], Auto-FedRL [8], and more.

**Federated Statistics**  FLARE provides built-in federated statistics operators (*Controller* and *Executors*) that will generate global statistics based on local client statistics. Each client could have one or more datasets, such as "train" and "test" datasets. Each dataset may have many features. For each feature in the dataset, FLARE will calculate the statistics and combine them to produce global statistics for all the numeric features. The output gathered on the server will be the complete statistics for all datasets in clients and global, as illustrated in Fig. 5.

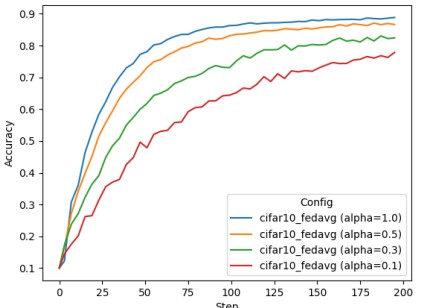 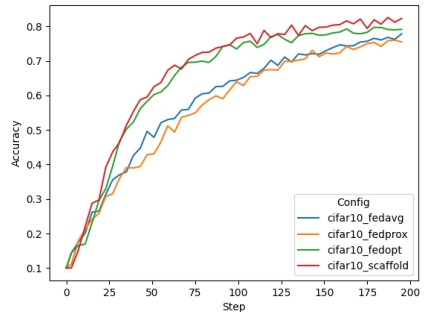

(a) FedAvg with increasing levels of heterogeneity (smaller $\alpha$ values).

(b) FL algorithms with a heterogeneous data split ($\alpha$=0.1).

Figure 4: Federated learning experiments with FLARE.

| | count | histogram |
|---|---|---|
| site-2-train | 6012 | [[0.0, 1.0, 12430030], [1.0, 2.0, 3511491], [2... |
| site-4-train | 1345 | NaN |
| site-3-train | 10192 | [[0.0, 1.0, 30867349], [1.0, 2.0, 4553187], [2... |
| site-1-train | 3616 | [[0.0, 1.0, 9512374], [1.0, 2.0, 1381654], [2.... |
| Global-train | 21165 | [[0.0, 1.0, 52809753], [1.0, 2.0, 9446332], [2... |

(a) Federated statistics. Note, the data of "site-4" violates the client's privacy policy and therefore does not share its statistics with the server.

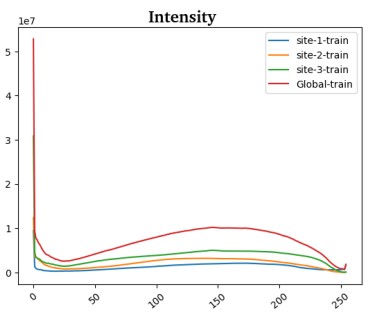

(b) Histogram visualization.

Figure 5: Federated statistics with FLARE.

**Real-world Use Cases** FLARE and its predecessors have been used in several real-world studies exploring FL for healthcare scenarios. The collaborations between multinational institutions tested and validated the utility of federated learning, pushing the envelope for training robust, generalizable AI models. These initiatives included FL for breast mammography classification [23], prostate segmentation [24], pancreas segmentation [28], and most recently, chest X-ray (CXR) and electronic health record (EHR) analysis to predict the oxygen requirement for patients arriving in the emergency department with symptoms of COVID-19 [6].

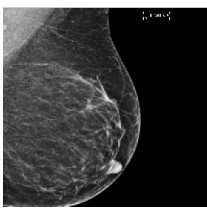 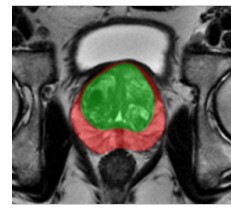 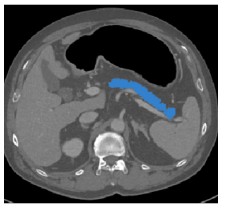 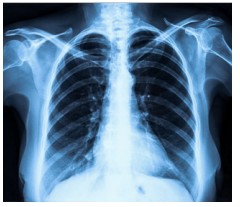

(a) Mammography.  (b) Prostate.  (c) Pancreas.  (d) CXR & EHR.

Figure 6: Real-world use cases of FLARE.

## 4 Summary & Conclusion

We described NVIDIA FLARE, an open-source SDK to make it easier for data scientists to use FL in their research and to allow an easy transition from research to real-world deployment. As discussed above, FLARE's *Controller* programming API supports various interaction patterns between the server and clients over internet connections, which could be unstable. Therefore, the API design mitigates

various failure conditions and unexpected crashes of the client machines such as allowing developers to process timeout conditions properly. FLARE is an open-source project. We invite the community to contribute and grow FLARE.

We did not go into all details of exciting features available in FLARE, like homomorphic encryption, TensorBoard streaming, provisioning web dashboard, integration with MONAI[4] [19, 3], etc. However, we hope that this overview of FLARE gives a good starting point for developers and researchers on their journey to using FL and federated data science in simulation and in the real world. For more information, please visit the code repository at `https://github.com/NVIDIA/NVFlare`.

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
