# OpenReview forum: "NVIDIA FLARE: Federated Learning from Simulation to Real-World"
_NeurIPS.cc/2022/Workshop/Federated_Learning — FL-NeurIPS 2022 Poster_

### Official Review · Reviewer_mmHi · 2022-10-11
**The advantages of FLARE compared to existing SDKs are not clear.**

1. This paper proposes a SDK, FLARE, to make it easier for data scientists to use FL in their research and real-world applications. The SDK includes solutions for state-of-the-art FL algorithms to facilitate building workflows for distributed learning across enterprises and enable platform developers to create a secure, privacy-preserving offering for multiparty collaboration utilizing homomorphic encryption or differential privacy.
2. Strengthens: It captures both simulated FL settings for researchers and prioritize production settings. In addition, it incorporates many exiting aggregation methods into the SDK and deploys it in real FL healthcare scenarios.
3. Weakness: It is not clear what are the highlights of this platform are not clear. Since there are so many open-source FL platforms for researchers and commercial purposes. Why should I choose FALRE? Like FederatedScope, one of its highlight might be the implementation of graph neural networks in FL.

---

### Official Review · Reviewer_Y7Cb · 2022-10-14
**A good overview of the proposed FL system, but lacking a few system metrics**

In this paper, the authors discuss FLARE, a newly developed federated learning framework mainly for cross-silo FL applications. They provide an overview of the functionality and modules included in the FLARE system. They also provide some numerical results in terms of FL model performance trained via FLARE on the CIFAR10 dataset. However, no system-related metric is provided for FLARE, for example, the runtime requirements and communication requirements with or without filter modules. Therefore, it is unclear what are the actual system performance and failure tolerance of FLARE.

---

### Official Review · Reviewer_cyMh · 2022-10-15
**In this paper, the authors introduced a novel open-source software development for the implementation of various FL.**

The following issues need to be solved:
1. Since the authors focused on the design of SDK, the technical contributions of this work are limited.
2.  There are several existing platforms for FL. The authors should compare the proposed SDK with existing platforms and point out the differences between them.
3. The authors should also compare the proposed SDK with existing platforms in simulations.

---

### Decision · Program_Chairs · 2022-10-20

Accept (Poster)